# Variation in Clinical Application of Hyperthermic Intraperitoneal Chemotherapy: A Review

**DOI:** 10.3390/cancers11010078

**Published:** 2019-01-11

**Authors:** Roxan F. C. P. A. Helderman, Daan R. Löke, H. Petra Kok, Arlene L. Oei, Pieter J. Tanis, Nicolaas A. P. Klaas Franken, Johannes Crezee

**Affiliations:** 1Department for Experimental Oncology and Radiobiology (LEXOR), Center for Experimental and Molecular and Molecular Medicine (CEMM), Amsterdam UMC, University of Amsterdam, Cancer Center Amsterdam, Meibergdreef, 1105 AZ Amsterdam, The Netherlands; f.c.helderman@amc.uva.nl (R.F.C.P.A.H.); a.l.oei@amc.uva.nl (A.L.O.); n.a.franken@amc.uva.nl (N.A.P.K.F.); 2Department of Radiation Oncology, Amsterdam UMC, University of Amsterdam, Meibergdreef, 1105 AZ Amsterdam, The Netherlands; h.p.kok@amc.uva.nl (H.P.K.); h.crezee@amc.uva.nl (J.C.); 3Department for Surgery, Amsterdam UMC, University of Amsterdam, Cancer Center Amsterdam, Meibergdreef, 1105 AZ Amsterdam, The Netherlands; p.j.tanis@amc.uva.nl

**Keywords:** peritoneal carcinomatosis, peritoneal metastasis, cytoreductive surgery, HIPEC, colorectal cancer (CRC)

## Abstract

Peritoneal metastasis (PM) originating from gastrointestinal and gynecological malignancies are associated with a poor prognosis and rapid disease progression. Cytoreductive surgery (CRS) with hyperthermic intraperitoneal chemotherapy (HIPEC) is an effective treatment option with curative intent. Hyperthermia enhances the cytotoxicity of chemotherapeutic drugs, thereby killing microscopic tumors and reducing the risk of tumor recurrence. Eight parameters potentially have an impact on the efficacy of HIPEC: the type of drug, drug concentrations, carrier solution, volume of the perfusate, temperature of the perfusate, duration of the treatment, the technique of delivery, and patient selection. In this review, a literature search was performed on PubMed, and a total of 564 articles were screened of which 168 articles were included. Although HIPEC is a successful treatment, there is no standardized method for delivering HIPEC: the choice of parameters is presently largely determined by institutional preferences. We discuss the current choice of the parameters and hypothesize about improvements toward uniform standardization. Quantifying the effect of each parameter separately is necessary to determine the optimal way to perform HIPEC procedures. In vivo, in vitro, in silico, and other experimental studies should shed light on the role of each of the eight parameters.

## 1. Introduction

Peritoneal surface malignancies are generally associated with poor prognosis and rapid disease progression. A major part of peritoneal surface malignancies are peritoneal metastases, also referred to as peritoneal carcinomatosis (PC), originating from several gastrointestinal and gynecological malignancies. Peritoneal metastasis (PM) is categorized as a metastatic and loco-regional disease limited to the abdominal cavity [1]. In most cancers, including colorectal cancer (CRC), metastases are the main cause of cancer-related deaths [2]. Approximately 5–10% of patients diagnosed with CRC are additionally diagnosed with PM, and in recurrent disease, the incidence increases to 20–60% [3,4]. PM of CRC origin is associated with poor prognosis, averaging only 5–24 months [5,6]. Patients with PM of gastric origin have poor prognosis, with median overall survival (OS) of 4–8 months, and a 5-year survival rate of approximately 3–6% [7,8]. Patients with PM originating from ovarian cancer have a 5-year survival rate of 25–29%, compared to a 5-year survival rate of >90% in early-stage ovarian cancer [9]. Besides the common gastrointestinal and gynecological malignancies, pseudomyxoma peritonei (PMP) and malignant peritoneal mesothelioma (MPM) are both rare peritoneal surface malignancies arising from primary mucinous tumors of the appendix and from the pleura, peritoneum, pericardium, and the tunica vaginalis testes, respectively [10,11,12]. A treatment option for these patients is cytoreductive surgery (CRS), but the eradication of all macroscopic visible tumors alone is likely not to be sufficient. Remaining microscopic disease can result in recurrence. Systemic chemotherapy has limitations when malignancies spread to the peritoneum due to the poor blood supply and low penetration into peritoneal tumor deposits. Therefore, in current practice, the treatment used for peritoneal metastasis from various origins is, increasingly, CRS, in combination with intraperitoneal chemotherapy (IPC). The combination of CRS and IPC changed the management of peritoneal malignancies from a palliative approach to a treatment with curative intent. There are two main ways of delivering IPC, namely early post-operative intraperitoneal chemotherapy (EPIC) and hyperthermic intraperitoneal chemotherapy (HIPEC). The former is used to perfuse the peritoneum on the first five days post-surgery via a catheter placed in the abdomen near the location where the risk of recurrence is largest [13]. During HIPEC, IPC is combined with hyperthermia (i.e., heating tumor tissue to 40–43 °C), increasing the lethality of selected drugs by enhanced cytotoxicity [14]. Right after CRS, a heated chemotherapeutic solution is circulated through the peritoneum for a maximum of 120 min. Hyperthermia is a treatment modality used in combination with surgery, radiotherapy, and chemotherapy [15]. Hyperthermia has a direct cytotoxic effect on cells in hypoxic, nutrient-deprived, and low-pH environments especially encountered in malignant tumor cells [16]. Hyperthermia affects the plasma membrane protein distribution influencing the membrane permeability [17] and the modulation of the transmembrane efflux pumps, enhancing the cytotoxic effects. Intracellular proteins are also affected, leading to impaired DNA repair, the denaturation of proteins, and the inhibition of repair enzymes, but also to the induction of heat-shock proteins [18,19].

Both CRS and HIPEC are beyond their respective pioneering phases. CRS was already used in combination with X-ray therapy for ovarian carcinoma patients by Meigs in the 1930s [20]. The development of HIPEC came almost half a century later, when Charles and Spratt designed the first “system for hyperthermic intracavitary perfusate” in 1977, two years before treating the first PMP patient in 1979 [21,22]. This technique was extended towards PM as a result from gastro-intestinal malignancies by Sugarbaker in the 1980s [20]. This was also the period in which the first phase 1 trials were undertaken. These trials were designed to investigate the pharmacokinetic advantages of intra-peritoneal drug delivery over intravenous drug delivery. The first trial was conducted in 1987, documenting the antineoplastic activity of cisplatin and etoposide [23]. From thereon, numerous studies and trials have been performed, trying to improve and optimize these techniques. The interest in HIPEC is sharply increasing in recent years because of the wider acceptance of this previously controversial technique. Figure 1 shows the number of HIPEC publications over the years. Over the last few decades, an exponential increase has been observed. However, the recently presented results of the multicenter randomized French Prodige-7 trial restarted the discussion of the additive effect of HIPEC to CRS in patients with PM from a CRC origin, especially following induction systemic therapy and using the 30 min high-dose oxaliplatin schedule for HIPEC [24]. However, at the same time, the multicenter randomized Dutch ovarian cancer (OV)HIPEC study showed a significantly improved OS with the addition of HIPEC to CRS in ovarian cancer [25].

A number of treatment parameters determine the efficacy of HIPEC. Kusamura [26] distinguishes seven parameters which are likely associated with the efficacy of HIPEC: the type of drug, drug concentrations, carrier solution, volume of the perfusate, temperature of the perfusate, treatment duration, and the technique of delivery (Figure 2) [21]. Up to now, there has been no standardized method for delivering HIPEC, with high variability among the different diseases and between institutes worldwide. In this article, we present an overview of the recent HIPEC literature and discuss these (seven) parameters of HIPEC, and argue that there is an eighth parameter influencing the efficacy of HIPEC—“patient selection”.

## 2. Methods

A literature search was performed on PubMed in August 2018. The search terms used were “HIPEC”, “Clinical Study”, “Colorectal Cancer”, “Ovarian Cancer”, “Gastric Cancer”, “Malignant Peritoneal Mesothelioma”, or “Pseudomyxoma Peritonei”. Only clinical research articles written in English and available in full text were included. These articles were screened on the complete reporting of the following parameters: drugs, dosage, temperature, duration, and delivery method. In total, 564 articles were screened, of which 168 articles were included in this study (Figure 3).

## 3. Parameters of HIPEC

### 3.1. Chemotherapy

The chemotherapeutic agents used in HIPEC procedures are selected based on several drug-specific characteristics. Cell-cycle specific chemotherapeutics, such as 5-fluorouracil (5-FU) and taxanes, are unfavorable as they require a long exposure time to induce cell death [27,28]. HIPEC procedures usually take between 30–120 min, a relatively short period in which cell-cycle non-specific chemotherapeutic agents need to achieve sufficient cell death [27]. The elevated temperature in the perfusate motivates the use of a type of chemotherapy which is synergistic with heat [29]. A large molecular size prevents quick absorption by the peritoneal surface, limiting systemic toxicity. The penetration depth of the drugs is generally in the order of a few millimeters, emphasizing the importance of a complete CRS to ensure only nodules smaller than 2.5 mm remain [27].

The most commonly used drugs for HIPEC are mitomycin C (MMC) and platinum-based cytotoxic drugs, including oxaliplatin, cisplatin, and carboplatin [23,24], which are all synergistic with heat. Less frequently used drugs are doxorubicin, irinotecan, docetaxel, paclitaxel, and 5-FU, where only the first is synergistic with heat. A high area-under-the-curve (AUC) ratio demonstrates limited systemic toxicity, associated with pooling. This is in combination with the cell-cycle-specific character which makes these drugs more suitable for EPIC. Table 1 summarizes the commonly used chemotherapy agents and their characteristics. An overview of the drugs used in different HIPEC clinics is shown the second and third columns of Appendix A.

### 3.2. Carrier Solution

The type of carrier solution plays a key role in the pharmacokinetics of HIPEC. Two core factors play a crucial role; the tonicity of the solution, and the molecular size of the solute. Tonicity is defined as the relative concentration of a solute between the interior and exterior of a cell. A relatively higher concentration inside the cell can result in an osmotic pressure gradient across the cell membrane, resulting in diffusion towards the exterior of the cell, lowering intracellular fluid volume and thus reducing the effective surface of the peritoneal cells. These kinds of solutions are known as hypertonic (Table 2). Its counterpart is a hypotonic solution, causing diffusion towards the interior of the cell, increasing the intracellular fluid volume and effective surface (Table 2). When the concentrations inside and outside of the cell are about equal, it is referred to as an isotonic concentration. The molecular size of the solutes in the solution can also influence the clearance from the peritoneal cavity into the cell and from the cell to the blood plasma. A high molecular weight and/or hypertonicity causes a slower clearance of peritoneal fluid, resulting in a prolonged exposure of chemotherapy to the peritoneal surface [30]. Other important aspects of the choice for a specific carrier solution is the influence of the stability of the chemotherapy and possible adverse effects. For example, oxaliplatin is known to become unstable in chloride-containing media. A degradation around 10% can be expected within 30 min and around 20% after 120 min when solved in a 0.9% NaCl solution [31]. For that reason, dextrose is often used in combination with oxaliplatin-based HIPECs. However, the use of dextrose solutions is linked to adverse effects. Long exposure to dextrose solution, both at low (1.5%, hypertonic) and high (5%, isotonic) concentrations, can cause hyperglycemia and serious disturbances in electrolyte concentrations [31,32].

The ideal carrier solution for HIPEC should improve exposure of the peritoneal surface, maintain high intraperitoneal volume, show slow clearance from the peritoneum, and not cause adverse effects to peritoneal membranes [33]. This improves the distribution of the drug and the efficacy of HIPEC. The carrier solutions used in HIPEC centers are shown in the third column of Appendix A.

### 3.3. Dosage and Perfusate Volume

The calculation of the drug dosage is based on the patient’s body surface determined in mg/m^2^, as variable per drug. The carrier solution volume (L/m^2^) is based on the patient’s body surface area, or an absolute volume is used. An often-used perfusate volume is 2 L/m^2^ [34,35]. The dosage and volume used in HIPEC centres are shown in the second and fifth columns of Appendix A. Depending on the institute, the drugs are administered in parts, or entirely at the start of perfusion. The drugs are given at once [36], but some protocols add ½ of the drugs at the start of a 90 min perfusion period, followed by ¼ of the drugs after 30 and 60 min [37,38,39]. The drug schedule used in HIPEC centres is in the fourth column of Appendix A.

### 3.4. Temperature

The synergism between cytotoxic drugs and hyperthermia starts at 39 °C and is stronger at higher temperatures, which is shown in several in vitro studies [14,40]. Heat reduces interstitial pressure and increases permeability, allowing better penetration of the drugs into the tumor cells and leading to apoptosis [41]. Moderate hyperthermia (41–43 °C) significantly enhanced the effect of cytotoxic drugs. Temperatures above 43 °C showed an even more intense effect, but also caused significant cytotoxicity in normal cells and are thus unsuitable for clinical use [14,42]. Table 3 summarizes the different types of hyperthermia used and their characteristics. An in vivo study performed by Shimizu et al. assessing thermo-tolerance concluded that 44 °C during 30 min was the maximum well-tolerated temperature in rats [42]. Therefore, the optimal temperature for HIPEC would be in the range of 40–43 °C.

During HIPEC treatments, temperatures are monitored, usually in the left-subphrenic area, right-subphrenic area, pelvic area near the outflow drains, and in or near the inflow catheter [33]. In the analyzed studies, the maintained intra-abdominal temperatures ranged from 39–44 °C. The temperatures used in HIPEC centres worldwide can be found in the seventh column of Appendix A.

### 3.5. Duration

Another parameter which influences the efficacy of HIPEC is the duration of perfusion. An in vitro study by Murata et al. showed that a longer exposure than 30 min to 5-FU or MMC under hyperthermic conditions did not significantly decrease the tumor cell survival rate, compared to a 30 min exposure or less [43]. However, cisplatin did show significantly less tumor cell survival rates after 60 min, compared to 30 min exposure under the same conditions. Unfortunately, there is currently no systematic study assessing the optimal duration of HIPEC in human or animal models. Overall, times ranging from 30–120 min was observed in the analyzed studies outlined in the eighth column in Appendix A.

### 3.6. HIPEC Delivery Techniques

HIPEC treatment can be delivered via different techniques: closed, open, semi-open, peritoneal cavity expander (PCE), and laparoscopic. The closed abdominal technique was the first technique of HIPEC, and approximately 50% is still being used in most institutes. After CRS, incisions are made in the flank of the abdomen for the placement of Tenckhoff catheters, temperature probes, and drains. Some institutes choose to place the drains through the long midline incision already used for the CRS. The abdomen is closed and filled with the hot perfusate to create a closed flow system. Preferably, reconstruction surgery is performed after perfusion to reduce the risk of recurrence along suture lines. Closing the abdomen makes it easier to obtain the required temperature for hyperthermia because of the reduced heat dissipation, while limiting the risk of theater staff from coming in contact with the chemotherapy. During treatment, the abdomen is agitated to improve homogeneity of the temperature and chemotherapy distribution. The perfusate distends the abdomen, but not enough to expand the space between the tightly packed peritoneal surface, resulting in an inhomogeneous distribution of the heated chemotherapy in the abdomen and leading to temperature differences. Experiments involving HIPEC treatments with methylene blue show exactly these inhomogeneities [13,44,45], a result from poor circulation. This can result in pooling, accumulating the heated chemotherapy in certain regions in the abdomen, thereby increasing the risks of systemic toxicity and heat-related injuries. Additionally, some areas remain untreated, possibly increasing the risk of recurrences. 

If a surgical team prefers to remain in contact with the interior of the abdomen, they might opt for the open technique. After CRS, separate incisions are made on the flanks of the abdomen for the placement of catheters, drains, and thermometry. The skin edges along the midline incision are attached to a Thompson retractor, creating an arena-like setup, also referred to as the “Coliseum” technique. A plastic sheet is placed over the midline incision, and a smoke evacuator is placed under the plastic sheet to reduce aerosolization of chemotherapy. See Figure 4 for a schematic figure of the Coliseum technique.

There are several advantages of the possibility to have surgical intervention during HIPEC. First of all, the surgeon can manually and slowly stir the abdominal contents to improve the homogeneity of the heated chemotherapy. Hot- and/or cold spots appearing on the temperature monitors or even observed by the touch of the surgeon can be corrected. Extra care should be given to the areas with increased risk of recurrence. In case of bleeding during perfusion, adequate and swift measures should be taken to minimize the risk of complications. Concentration, temperature, and other intra-abdominal variables can be easily monitored during surgery, creating the opportunity to create optimal conditions for the HIPEC [46,47]. The aerosolization of chemotherapy is a relevant potential hazard during open-abdomen HIPEC, and is one of the core arguments for using a closed procedure. However, extensive tests involving urine and blood samples of surgical personnel, air samples from the operating room, and penetrability of surgical gloves showed no sign of additional health risks for the operating staff [48].

The advantages and disadvantages of closed and open techniques motivated the development of semi-open techniques, where the abdomen is open during surgical interventions but effectively closed during the rest of the perfusion, decreasing heat loss and the possible spread of aerosols. The exact design differs per institute; one example is by placing a lid or a semi-penetrable hole in the middle of a thick non-permeable sheet covering the abdomen [46]. More exotic techniques involve a PCE. Several implementations exist, but the first attempt dates from 1990 when Fujimura et al. [49] created an acrylics-made cylinder with a spindle-shaped cross-section placed in the peritoneum, enabling the small bowel to float freely in the cylinder, spatially above and outside the abdomen. Rat et al. [50] used an expander by stapling it watertight to the abdomen and a latex sheet. The latex sheet served as an overflow region for the chemotherapy solution. Like the semi-open techniques, it featured a “glove-box” type of entry to the abdomen. Expanding the total volume of the peritoneum creates an environment where the effective peritoneal and visceral surface is enlarged. The expanded incision enhances the accessibility. In the closed and open techniques, organs are packed more tightly, reducing the effective surface and limiting the level of homogeneity reached. The agitation and manipulation of the abdomen and its contents does improve uniformity, but homogeneity over the entire peritoneal surface is difficult to achieve. The expander covers a part of the peritoneal wound, shielding that region from the heated chemotherapy, leaving it untreated and thereby risking possible recurrences along that area.

In recent years, laparoscopic approaches have been developed. The main advantage of this approach is its minimal invasiveness, in contrast to conventional HIPEC techniques. These laparoscopic approaches are used in different settings. If the tumor burden is small (based on the peritoneal cancer index (PCI) score), some groups prefer to perform HIPEC and CRS using a laparoscopic approach. However, there is a significant risk of understaging with underestimation of the PCI, because not all areas of the abdominal cavity can be adequately explored using a laparoscopic approach. If an extensive tumor load is observed during the diagnostic laparoscopy, more conventional techniques are used instead [51]. There are also cases in which a laparoscopic HIPEC can be used in a palliative [52] or in a preventive setting [53]. An important observation is that a laparoscopic approach is effectively closed and disadvantages observed in the closed technique are to be expected to occur in laparoscopic approaches as well. Laparoscopic HIPEC has been associated with better penetration of the chemotherapy because of increased intra-abdominal pressure [54].

All advantages and disadvantages regarding the choice of technique are listed in Table 4. Despite the different advantages and disadvantages of the delivery techniques, the clinical outcome in terms of survival and morbidity rates is similar [55,56,57]. Differences in intraoperative parameters, such as blood pressure, pulse rate, core temperature, etc., are also not statistically significant between the techniques [55]. This absence of significant differences is reflected in a lack of consensus on the preferred technique. To the best of our knowledge, there is not enough evidence to determine the superiority of one technique over the other [13]. In the analyzed studies, 53% of the institutes used the closed technique, the open technique was used in 42%, and only 5% used the semi-open, PCE, or laparoscopic techniques. The ninth column of Appendix A shows details on the delivery technique.

### 3.7. Patient Selection

Patient selection for CRS and HIPEC is very important, because it has a major impact on the chance of a positive outcome, and not every patient is suitable for HIPEC [58]. Patients should be in good clinical condition, and selected based on the extent of peritoneal disease, presence of distant metastasis, estimated completeness of cytoreduction, histological subtype, and disease origin [59,60].

Prior to surgery, the extent to which different peritoneal sites are involved is determined based on tumor size and location according to the PCI. The PCI, postulated by Sugarbaker in 1996, can be non-invasively assessed by computed tomography (CT). This is often followed by diagnostic laparoscopy due to the inaccuracy of CT, but is increasingly being replaced by MRI [51,52]. The abdomen is segmented into 13 regions, each of which are scored, ranging from 0 (no tumor) to 3 (tumors larger than 5 cm). The scores per region are added, resulting in a PCI between 0 and 39. The PCI has important prognostic implications, and it also provides an indication of the chance of performing a complete CRS. A more recently developed scoring system, the Peritoneal Surface Disease Severity Score (PSDSS), extends the PCI score by taking the symptoms of the patient and the primary tumor histopathology into account [61,62]. The PCI/PSDSS continue to have a fundamental role in patient selection.

The completeness of the CRS, based on residual tumor sizes, is an essential prognostic variable. Complete cytoreduction (CCR), scored as CCR-0, is defined as the eradication of all macroscopic nodules. Incomplete cytoreduction is partitioned into two categories: CCR-1 and CCR-2. In the latter, large lesions are still present in the abdomen (usually >2.5 mm), but this varies [18]. In the former, macroscopic lesions smaller than 2.5 mm remain. Multivariate analysis show that a CCR-0 score is key for long-term survival [16,19].

In CRC, only approximately 25% of the patients diagnosed with PM are eligible for the combination of CRS and HIPEC. Besides the PCI and CCR scoring, several other patient selection criteria are used. The condition of the patient is assessed by using the performance status and Karnofsky index. Both criteria measure the level of functioning. Other criteria used are metastatic extent and lymph node involvement. Usually, only patients with a performance status of ≤2, Karnofsky index of >70, PCI-score of <20, absence of distant metastasis, no lymph node involvement, and complete or partially resectable peritoneal disease (CCR-0/1) are considered suitable for CRS and HIPEC. Patient selection criteria used for HIPEC are listed in Table 5.

### 3.8. Additional Parameters

As there is no consensus on the optimal technique, the efficacy of HIPEC can also potentially be improved by optimizing technique-independent variables. Besides the parameters discussed above, there are two additional parameters that are important for the quality of HIPEC treatment—namely, the interstitial fluid pressure and the flow rate. The interstitial fluid pressure in tumors is known to be a therapeutic barrier [63]. Increasing intra-abdominal pressure reduces the pressure gradient between the interior and exterior of peritoneal nodules. This magnifies the effect of the interstitial pressure reduction caused by hyperthermia itself. Facy et al. [54] studied the effect of intra-abdominal pressure by applying a water column during open-abdomen HIPEC. This study differentiated between four groups, varying the temperature (38 °C or 42 °C) and pressure (atmospheric and hydrostatic, applied by a 25 cm water column). It was found that the highest tissue concentrations were found in the hyperthermic and hyperbaric group [54]. This conclusion confirmed results found by [64,65], using maximal intra-abdominal pressures of 30 mm Hg and 40 mm Hg, respectively. Complications such as renal failures due to high pressures [64] and ischemia in the viscera after long treatments (60 min) were observed [65]. Application of moderate pressure, maybe area-dependent, could be clinically relevant.

Maintaining high flow rates during HIPEC is important in achieving and maintaining the required temperature for hyperthermia. Clinical flow rates are generally between 0.5–2 L per minute (Appendix A). Furman et al. [66] explored the influence of flow rate on the heating quality of the peritoneal cavity. Four flow-rate regimes (notably 1, 2, 3, and 4 L per minute) were investigated in a water tank with a suspended saline bag acting as viscera. Higher flow rates resulted in a more rapid heating of the compartment modelling as peritoneum [66]. This could possibly spare the interior of the viscera due to a strong temperature gradient along the depth of the viscera. However, there could be adverse effects, due to excessively high flow rates and rapid heating. One also has to consider the number and location of inflow catheters. A focused flow directed at a particular surface for a large portion of the duration of perfusion could cause thermal injuries at that location. Maintaining a rapid flow does not always imply that a homogeneous chemotherapy distribution is realized in all parts of the peritoneum. Flow rate, and the placement of the catheters and drains, in combination with the technique of delivery are important factors for obtaining homogeneity. Flow rates described in the literature are shown in the sixth column of Appendix A.

## 4. Discussion

HIPEC, in combination with CRS, has been shown to improve the oncological outcome of selected patients with peritoneal malignancies to such an extent that the treatment can be given with curative intent. This improvement is underlined in the median OS column of Appendix A, although it is highly dependent on the underlying primary disease. For CRC patients, the average median OS was observed to be 32.3 months, ranging between 13 and 76.9 months, and with an average 5-year survival of 44.5%, ranging from 19 to 83.3 months for the studies included in this analysis. Systemic chemotherapy has also improved over recent years with a median OS of 13 months, ranging between 5 and 24 months, and a 5-year survival rate between 0 and 22% [6]. When ovarian cancer was the primary cause of PM, we observed an average median OS of 39.2 months, ranging between 5.8 and 65.6 months, and an average 5-year survival rate of 43.9%, ranging between 12 and 63.4%. Systemic therapies show a median OS ranging between 3.5 and 62 months, with a 5-year survival rate below 25% [67]. Patients suffering from PM from gastric origin have the grimmest prognosis, with median OS ranging from 4–8 months [8] and a 5-year survival rate between 3–6% [7]. Studies included in this analysis reported an average median OS of 17 months, ranging between 7.8 and 60.85 months and an average 5-year survival rate of 39%, ranging between 0 and 83.3%.

Considering the wide ranges observed in the included studies, a large number of patients gain no advantage from their treatment. These patients underwent extensive and invasive surgery while not gaining a clear survival benefit over the patients receiving palliative treatment. The cause for this discrepancy in survival between institutes/studies should be brought to light, and further steps should be taken to optimize patient selection with more tailored approaches. Standardization can be key for providing equal care for all patients, and the eight parameters discussed in this article should provide a guideline for delimiting the underlying cause of treatment failure.

In Figure 5, the variability between institutes for six of the eight different parameters is summarized. Note that patient selection and dose are not included in these figures, since these are not easily categorized. Drugs (Figure 5a) most often used in HIPEC treatments are MMC and cisplatin, and both are observed to have enhanced lethality at elevated temperatures. Oxaliplatin, also synergetic with heat, is too often used in HIPEC treatments for patients with PM of CRC origin [66]. HIPEC, for ovarian cancer patients, is mostly performed with “other” drugs. This category embodies 5-FU, carboplatin, eloxatin, docetaxel, gemcitabine, laboplatin, and/or irinotecan. The lethality of a drug depends on the cancer origin and the temperature at which it is applied. This can be investigated with in vitro studies, which has already been addressed for various agents and cell lines [14,27,38,67]. In these in vitro studies, the thermal enhancement ratio (TER) of drugs are determined. TER is the ratio of the dose to achieve an endpoint at 37 °C, to the dose to achieve the same endpoint combined with an elevated temperature [68]. These studies have shown that the TER increases with an increase in the temperature. However, this data is not yet complete, and further experiments should be conducted in this field. The TERs should be determined within 0.5 °C to be able to reach a conclusion on the ideal temperature.

The prevalent choice of carrier solution is a saline solution or a dextrose solution. The different choices of carrier solution and their frequency can be seen in Figure 5b. Carrier solution, type of drugs, concentration, volume of the perfusate, and heat all have a great impact on the clearance of the drugs, from the peritoneal cavity to the plasma, and the way the drugs accumulate between cells. The tonicity and molecular weight are two key factors for choosing the optimal carrier solution. In vitro experiments can map the influence on the accumulation between cells and via in vivo experiments, and the influence on the drug/fluid clearance and plasma drug concentrations can be investigated [30,68].

The duration of HIPEC treatments varies between 30 min to 120 min, as observed in Figure 5c. In clinical practice, two opposing regimens are used: high temperature and/or dose for a short period of time (usually 30–45 min) versus low dose and/or temperature for a long period of time (usually 60–90 min). The influence of the duration on possible adverse effects, efficacy of the treatment, or possible immunity is not exactly known and is difficult to differentiate from other parameters, such as the type of drug, drug concentration, temperature, and carrier solution. After having a clear picture of the impact of these four parameters, it might be possible to determine the best duration through in vivo studies.

The volume of the perfusate is determined in two ways: based on an absolute volume varying between 1 and 12 L, or dependent on the patient’s body surface (often 2 L/m^2^). The frequency of these two choices can be seen in Figure 5e. It the latter case, the drug concentration is uniform over the patient, while this is not the case in the former. A limitation is that a patient’s body surface does not necessarily say anything about the peritoneal cavity volume of the patient. However, the concentration plays a key role, together with the duration, temperature, type of drug, carrier solution, and dose, in how much of the drug is actually delivered into the tumor cells and systemically absorbed. A stable concentration can limit variability, and should be preferred over an absolute volume.

As previously mentioned, the choice of temperature depends on the regimen used. Usually, high temperatures are combined with a low(er) dose and/or for a short(er) period, and low temperatures are combined with high(er) dose and/or long(er) duration. In Figure 6b, we present a world map representing the temperature used around the world. The temperature range is between 41 and 43 °C around the world, as observed in Figure 5d, as was determined to be the optimal temperature for HIPEC at the 2006 Milan convention [26]. This seems to be a very uniform parameter, but a difference of a few degrees can increase the lethality of a chemotherapeutic agent with a factor of 2–4 or, in some chemotherapies, even more [29]. The ideal temperature depends on the type of drug, the maximal temperature clinically tolerated, and the TER values of drugs at a certain temperature. Both in vitro and in vivo experiments can be employed to find these values [14,42]. In Figure 6a, we present a similar figure, but now showing the technique prevalent in that country. The closed technique remains as the most frequently used technique (also see Figure 5f). The countries lined in gold represent the countries where other techniques, such as the semi-open, PCE, or laparoscopic techniques are used as well. HIPEC is delivered as an independent parameter, and it only concerns the homogeneous delivery of the other parameters. Flow patterns and heat exchange determine these conditions, making this parameter very suitable to investigate in (flow) experimental set-ups and in silico studies.

An extensive statistical analysis including all patient characteristics can be useful to differentiate between the choice of parameters, but only when a sufficiently large patient cohort is used. Even then, it can become difficult to obtain statistically significant results because of the large amount of treatment parameters and large variability between parameters. For the same reason, we argue that randomized controlled trials are not an effective way to determine the contribution of each of these parameters. Experimental studies should be able to give the choice of parameters a strong scientific foundation without involving the patient. There is significant interdependence between the eight parameters we discussed. In Figure 7, we present a flow chart in which we hypothesize the correct way of determining the optimal HIPEC treatment for a specific patient. Below the parameter, we list the way in which we can determine this parameter. It should be emphasized that patient selection has a major impact on the outcome of HIPEC. Patient selection does not directly depend on the other parameters, and it only makes sense to perform these clinical studies if the other parameters are determined in full. The volume should be based on the patient’s body surface to ensure that the same molarity is obtained for every patient. The type of drug, concentration, carrier solution, duration, and temperature should be based on in vitro and in vivo studies, and are all co-dependent on each other. Determining the delivery technique should be based on in vivo, in silico, and experimental studies. Since it only concerns the homogenous delivery of the other five parameters, it can also be considered as an independent parameter.

## 5. Conclusions

In this review a literature search was performed on PubMed, and a total of 564 articles were screened, of which 168 articles were included. Eight parameters were found to have an impact on the efficacy of HIPEC: the type of drug, drug concentrations, carrier solution, volume of the perfusate, temperature of the perfusate, treatment duration, the technique of delivery, and patient selection. The variability in worldwide application of HIPEC considering each of the eight parameters is substantial, which might be one of the major determinants for the observed variation in efficacy of CRS/HIPEC. The eight parameters should be further explored to achieve a stronger scientific foundation, which is incomplete at the moment. Quantifying the effect of each parameter separately can help to optimize treatment protocols and thereby further improve the efficacy of HIPEC. In vivo, in vitro, in silico, and other experimental studies are valuable tools for determining the contribution of the individual parameters.

## Figures and Tables

**Figure 1 cancers-11-00078-f001:**
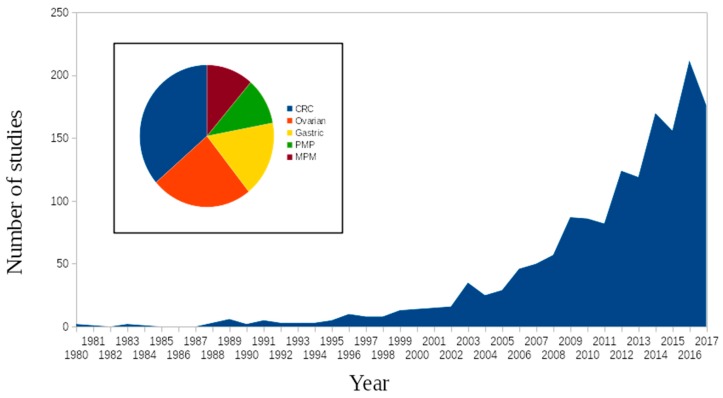
Number of studies published “hyperthermic” and “peritoneal” and “chemotherapy” on PubMed. Pie chart showing the distribution of each cancer origin of all included papers in this review.

**Figure 2 cancers-11-00078-f002:**
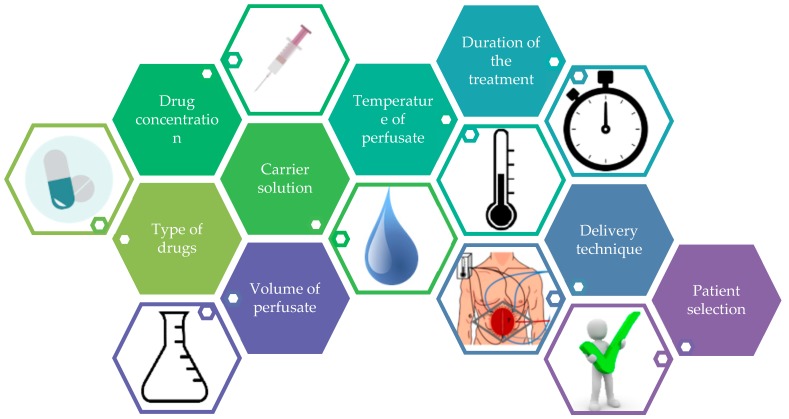
Eight parameters of hyperthermic intraperitoneal chemotherapy (HIPEC).

**Figure 3 cancers-11-00078-f003:**
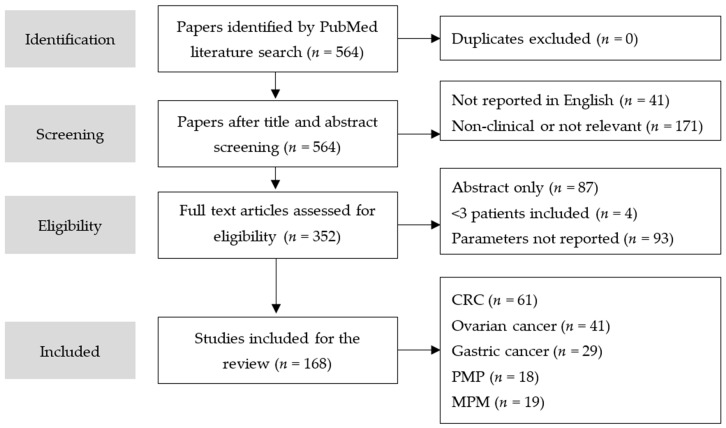
Flowchart literature study. CRC: 190 hits, of which 61 were included; Ovarian: 131, of which 41 were included; Gastric: 106, of which 29 were included, PMP: 46 hits, of which 18 were included; MPM: 91 hits, of which 19 were included.

**Figure 4 cancers-11-00078-f004:**
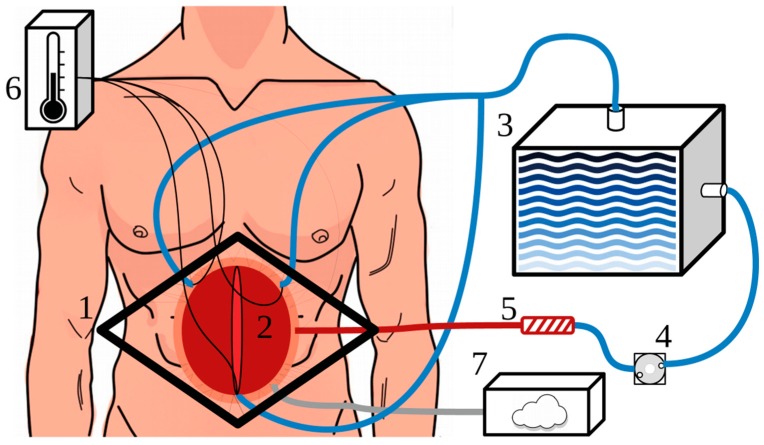
Schematic diagram of an open HIPEC procedure, with one inflow catheter and three outflow drains. The abdominal wall is retracted by sutures attached to a Thompson retractor (1). A plastic sheet is placed over the open abdomen (2) and an incision is made in the plastic to enable access to the abdominal contents. The heated carrier solution is stored in the reservoir (3) and is circulated by a roller pump (4). The inflow perfusate passes to a heat exchanger (5) to heat the solution to the required temperature. The temperatures are monitored during the treatment (6), usually in the left-subphrenic area, right-subphrenic area, pelvic area near the outflow drains, and in or near the inflow catheter. The smoke evacuator (7) is placed under the plastic sheet to prevent the aerosolization of chemotherapy.

**Figure 5 cancers-11-00078-f005:**
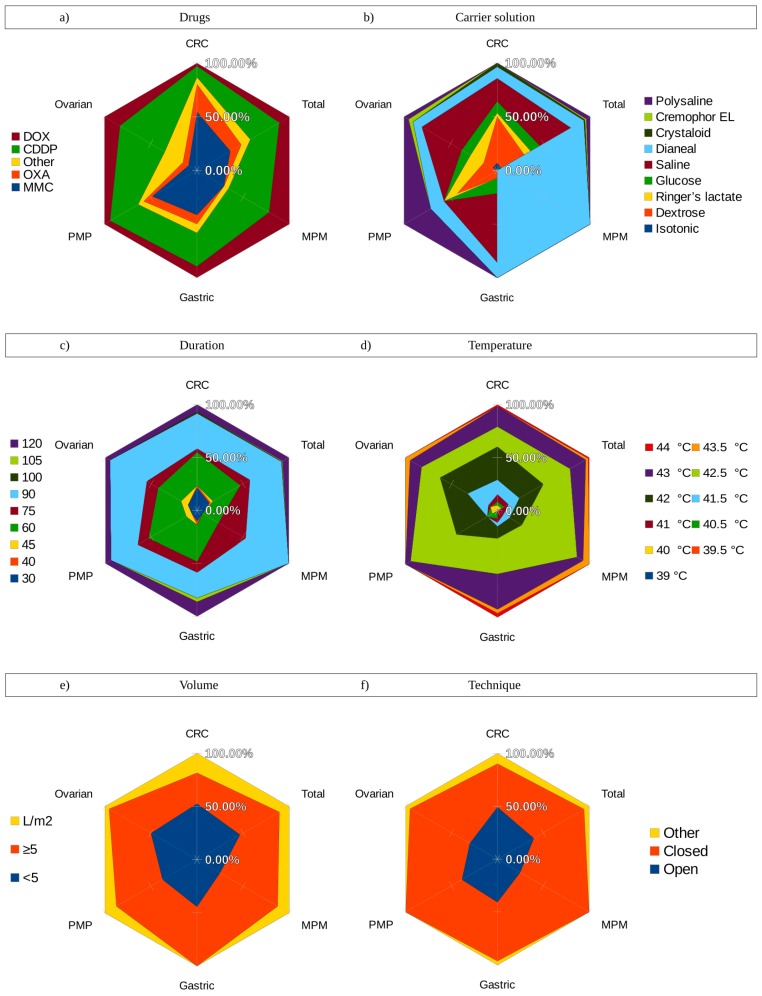
Data collected from the literature for six parameters influencing the efficacy of HIPEC: (**a**) Drugs; (**b**) carrier solution; (**c**) duration; (**d**) volume; (**e**) temperature; and (**f**) technique. CRC: colorectal cancer; PMP: pseudomyxoma peritonei; MPM: malignant peritoneal mesothelioma.

**Figure 6 cancers-11-00078-f006:**
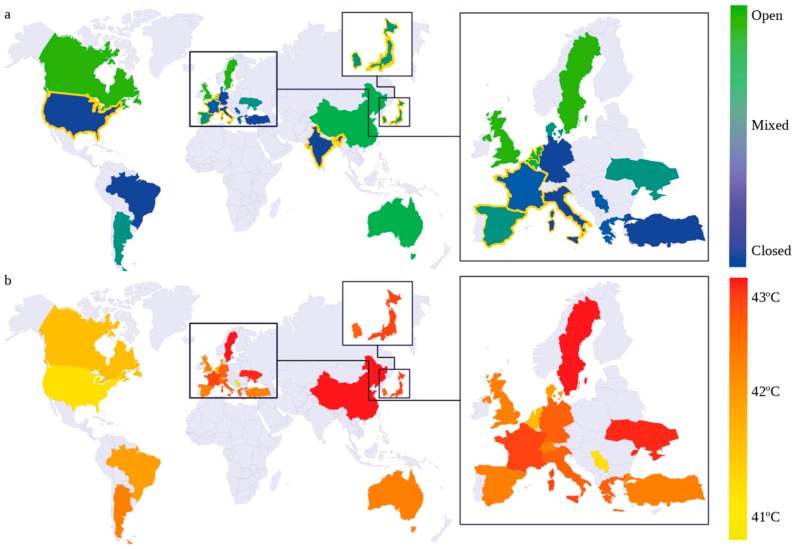
World maps showing the geographic variability of techniques (**a**) and median temperatures (**b**) used for HIPEC treatments. Countries where semi-open, PCE, or laparoscopic techniques are used are lined with gold.

**Figure 7 cancers-11-00078-f007:**
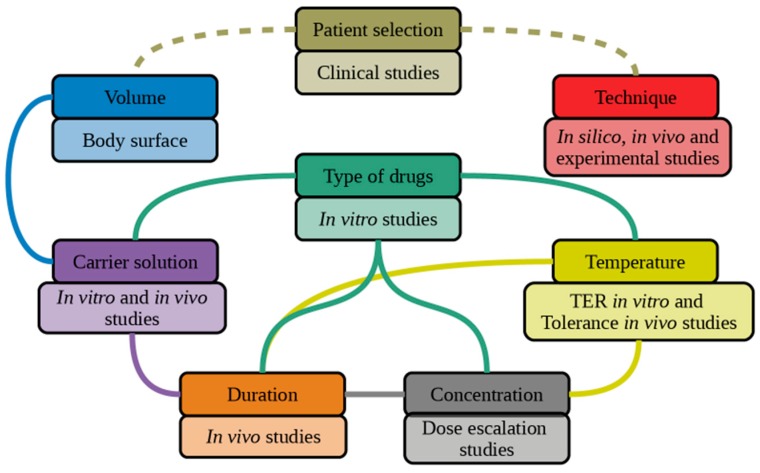
Flow diagram for the correct determination of treatment parameters.

**Table 1 cancers-11-00078-t001:** Cytotoxic drugs used for HIPEC (adapted from Kusamura et al., 2008 [26]). Dosage based on literature search (see Appendix A).

Drug	Type	Dosage (mg/m^2^)	AUC Ratio	Synergistic with Heat	Penetration Depth (mm)	Cell-Cycle Specific
Mitomycin C	Antitumor antibiotic (methylazirinopyrroloindoledione antineoplastic)	10–160	23.5	Yes	2	No
Oxaliplatin	Alkylating agent	160–460	16	Yes	1–2	No
Cisplatin	Alkylating agent	50–360	7.8	Yes	1–3	No
Doxorubicin	Antitumor antibiotic (anthracycline topoisomerase inhibitor)	15	230	Yes	4–6 cell layers	No
Irinotecan	Plant alkaloids (topoisomerase inhibitor)	100–400	N/A	No	N/A	Yes
Paclitaxel	Plant alkaloids (microtubule inhibitor)	60–175	1000	No	>80 cell layers	Yes
Docetaxel	Plant alkaloids (microtubule inhibitor)	80	552	No	N/A	Yes
5-fluorouracil	Antimetabolite (nucleoside metabolic inhibitor)	1000	250	Minimal	0.2	Yes
Carboplatin	Alkylating agent	350–800	10	Yes	0.5	No

^1^ AUC: Area Under the Curve; N/A: Not Available.

**Table 2 cancers-11-00078-t002:** Types of carrier solutions (adapted from Kusamura et al. (2008) [26]).

Type of Carrier Solution	Advantages	Disadvantages
Isotonic salt solutions and Dextrose solutions	Rapidly absorbed due to low molecular weight	Inability to maintain a prolonged high intraperitoneal fluid volume
Hypotonic solutions	Increases the cisplatin accumulation and enhances its cytotoxicity in vitro	Unexplained postoperative peritoneal bleeding
Hypertonic solutions	Allows prolonged high intraperitoneal volume Slows down the clearance of intraperitoneal fluid	Dilution of intraperitoneal drug due to fluid shift inward to the peritoneal cavity
Isotonic molecular weight solutions	Prolonged high intraperitoneal volume Reduced drug clearance from the peritoneal cavity	Drug exposure to the cancer cells is not significantly increased

**Table 3 cancers-11-00078-t003:** Temperatures used for HIPEC.

Temperature	Type of Hyperthermia	Cytotoxic Effect	Thermosensitization	Vascular Effect	Immune Reaction
39–41 °C	Mild	Minimal growth arrest	Synergism with cytotoxic drugs	Increased blood flow	Enhanced
31–43 °C	Moderate	Reversible growth arrest	Significant increased effect combined with cytotoxic drugs	Increased blood flow	Enhanced
>43 °C	Severe	Exponential growth arrest, significant cytotoxicity in normal cells	Significant increased effect combined with cytotoxic drugs	Reduced blood flow	Suppressed

**Table 4 cancers-11-00078-t004:** Comparison between the four different available techniques.

Type of Technique	Advantages	Disadvantages
“Colosseum” technique	Relatively uniform drug distribution Relatively uniform temperature distribution Surgical interceptions possible Manual creation of optimal conditions	Heat dissipation ^1^ Possible aerosolization of chemotherapy ^1^
Closed abdomen	Limited heat dissipation Reduced risk for theater staff	Minimal surgical interception possibilities Drug inhomogeneities
Peritoneal Cavity Expander (PCE)	Drug/heat distribution homogeneous Increased perfusion volume increasing effective surface Minimized exposure to theater staff Enlarged accessibility	Complex technique Experienced staff required
Laparoscopic	Limited heat dissipation Minimally invasive Can be used in different settings	Can only be used in combination with low tumor burden Drug and heat inhomogeneities

^1^ Semi-open approach improves the heat dissipation and reduces aerosolization.

**Table 5 cancers-11-00078-t005:** Patient selection criteria used for HIPEC.

Criteria	Factor	Inclusion Criteria	Details
Performance status	0 (able to carry out all normal activity) 4 (completely disabled)	≤2	Measuring a patient’s level of functioning in terms of their ability to care for themselves, daily activity, and physical ability
Karnofsky index	0 (dead/moribund) 100 (no evidence of Disease)	>70	A standard way of measuring the ability of cancer patients to perform ordinary tasks
PCI	0 (no tumor) 39 (tumors greater than 5 cm spread through the Peritoneum)	CRC + gastric cancer: PCI ≤ 10; MPM + PMP + ovarian cancer: no PCI limit	Extent of disease at the time of surgery
Metastatic extent	0 (no distant metastasis), 1 (distant metastasis present)	M0	Score to define distant metastasis
Lymph node involvement	0 (no involvement, 1 (1–3 node), 2 (>4 node)	N0 or N1/2	Score to define lymph node involvement metastasis
CCR Score	0 (complete CRS), 1 (0–2.5 mm), 2 (>2.5 mm)	CCR-0 or CCR-1/2	Completeness of cytoreduction score after cytoreduction to assess residual nodules

PCI: peritoneal cancer index; CCR: Complete cytoreduction; MPM: malignant peritoneal mesothelioma; PMP: pseudomyxoma peritonei.

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
