# Peer review of "Variation in Clinical Application of Hyperthermic Intraperitoneal Chemotherapy: A Review"

_cancers, 2019, doi:10.3390/cancers11010078_

Round 1

Reviewer 1 Report

The quality of figure 5 should be improved.

My main concern is why 87 papers were excluded from analysis due to no full text available - this is significant number of papers! 

Author Response

Comment: The quality of figure 5 should be improved.

Answer: Quality of figure 5 has been improved by increasing the size of each sub-diagram making it better readable and increasing the space for the legends.

Comment: My main concern is why 87 papers were excluded from analysis due to no full text available - this is significant number of papers!

Answer: Our apologies for the misunderstanding. All 87 papers were actually just abstracts, and not full papers. We rephrased this statement to reflect this.

Reviewer 2 Report

Well written review on HIPEC for several cancers with PM.

I only have some minor suggestions:

Tables:

- Please include a table or figure showing a summary of the results or the current state of HIPEC regarding the different cancers

- please reformat tables. It is very difficult to clearly see which information belongs to each line.

- table 1: antibiotic --> please specify

Table 1 and 2: I think, it is not optimal to include 2 tables which are mainly extractions from one already published review (kusamura et al). It would be better and more scientifically sound to include some new data. Please reconsider optimizing these tables.

Text:

Please check abbreviations. Any abbreviations should be explained when first mentioned!

Line 93: In this --> remove tab

Line 94: eight --> eighth

line 329: by who?????

Focus on patient selection is very important and could be more emphasized. What suggestions do the authors have for new studies clearly providing the evidence we Need?

Author Response

Comment: Please include a table or figure showing a summary of the results or the current state of HIPEC regarding the different cancers

Answer: We appreciate the value of such a summarizing table/figure, but we also feel that the results of HIPEC are nuanced and can therefore not be adequately summarized in a single table. We added a pie-chart to figure 2 representing the current state of HIPEC.

Comment: please reformat tables. It is very difficult to clearly see which information belongs to each line.

Answer: Done. We added shadings to improve readability.

Comment: table 1: antibiotic --> please specify & I think, it is not optimal to include 2 tables which are mainly extractions from one already published review (kusamura et al). It would be better and more scientifically sound to include some new data. Please reconsider optimizing these tables.

Answer: Added specifications in table 1. We understand your request but tables 1 and 2 (which were adapted from Kusamura) represents the opinion of a 2008 consensus meeting for the current values, and we did drastically rearrange and shorten the table as listed in the original paper. Dosage is the parameter we added to these tables from our own literature search. 

Comment: Please check abbreviations. Any abbreviations should be explained when first mentioned!

Answer: We have checked abbreviations and added explanations where necessary.

Comment: Line 93: In this --> remove tab, Line 94: eight --> eighth, line 329: by who?????

Answer: Lines 93, 94 and 329 have been adjusted accordingly.

Comment: Focus on patient selection is very important and could be more emphasized. What suggestions do the authors have for new studies clearly providing the evidence we Need?

Answer: The reviewer is right that patient selection is very important for clinical applications of HIPEC. We added a sentence emphasizing the importance of patient selection:

 “In figure 7, we present a flow chart in which we hypothesize the correct way of determining the optimal HIPEC treatment for a specific patient. Below the parameter we list the way in which we can determine this parameter. It should be emphasized that patient selection has a major impact on the outcome of HIPEC.. Patient selection does not directly depend on the other parameters and it only makes sense to perform these clinical studies if the other parameters are determined in full.”

To support selection and to establish clinical protocols for conducting HIPEC trials we recommend in vitro, in vivo and in silico studies for investigating the impact of various parameters.

Reviewer 3 Report

The paper is a nicely written detailed review that summarize the studies on the application of hyperthermic intraperitoneal chemotherapy for the treatment of metastatic cancer deriving from gastrointestinal and gynaecological malignant tumours. The survey includes a deep analysis of eight parameters that represent the major contributing factor in the efficacy variation of treatments (CRS/HIPEC).

The paper falls within the scope of the journal, the presented data are really interesting and worthy to be published. Anyway, the manuscript requires small changes before it can be accepted.

Keywords should not contain the same words present in the title, “Hyperthermic Intraperitoneal Chemotherapy” should be removed.

Paragraphs 3-7 should be merged into a single section with sub-paragraphs and the different factors taken into consideration should be described in a shorter way, moving a deeper description to the discussion section (paragraph 11).

Lines 196-198 should be rewritten more clearly avoiding repetitions of words (institutions). In general, there are some other repetitions, both of words and concepts, that make reading not smooth (i.e. “poor circulation. Poor circulation” line 210; “…laparoscopic approaches are being developed. Laparoscopic approaches...” line 251 and so on). This should be avoided.

If possible Table 5 (Patient selection criteria used for HIPEC) should be redesigned to be clearer.

The conclusions should be extended highlighting the key points of the survey and providing a more comprehensive summary of all studied data.

Bibliographic references should be formatted according to the journal rules.

After these modifications the paper could be accepted for publication.

Author Response

Comment: Keywords should not contain the same words present in the title, “Hyperthermic Intraperitoneal Chemotherapy” should be removed.

Answer: The keyword “Hyperthermic Intraperitoneal Chemotherapy” has been removed.

Comment: Paragraphs 3-7 should be merged into a single section with sub-paragraphs and the different factors taken into consideration should be described in a shorter way, moving a deeper description to the discussion section (paragraph 11).

Answer: We understand that the intended structure was not completely clear and have now amended this by introducing sub-headings representing each of the eight parameters.

Comment: Lines 196-198 should be rewritten more clearly avoiding repetitions of words (institutions). In general, there are some other repetitions, both of words and concepts, that make reading not smooth (i.e. “poor circulation. Poor circulation” line 210; “…laparoscopic approaches are being developed. Laparoscopic approaches...” line 251 and so on). This should be avoided.

Answer: We checked on repetitions and rephrased sentences to make them more easy to read.

Comment: If possible Table 5 (Patient selection criteria used for HIPEC) should be redesigned to be clearer.

Answer: To make table 5 more clear we have removed 1 column and added shadings.

Comment: The conclusions should be extended highlighting the key points of the survey and providing a more comprehensive summary of all studied data.

Answer: Yes you are right, we added the key points of the survey and summarized the studied parameters.

Comment: Bibliographic references should be formatted according to the journal rules.

Answer: References have been formatted according to the guidelines of the journal.